# Controllable Contrastive Generation for Multilingual Biomedical Entity Linking

**Tiantian Zhu[1], Yang Qin[1], Qingcai Chen[1,2,*], Xin Mu[2], Changlong Yu[3], Yang Xiang[2,*]**

[1]Harbin Institute of Technology (Shenzhen), Shenzhen, China
[2]Peng Cheng Laboratory, Shenzhen, China
[3]The Hong Kong University of Science and Technology, Hong Kong, China
zhu.tiantian110@gmail.com, {csyqin, qingcai.chen}@hit.edu.cn,
{mux, xiangy}@pcl.ac.cn, cyuaq@cse.ust.hk

## Abstract

Multilingual biomedical entity linking (MBEL) aims to map language-specific mentions in the biomedical text to standardized concepts in a multilingual knowledge base (KB) such as Unified Medical Language System (UMLS). In this paper, we propose **Con2GEN**, a prompt-based **con**trollable **con**trastive **gen**eration framework for MBEL, which summarizes multidimensional information of the UMLS concept mentioned in biomedical text into a natural sentence following a predefined template. Instead of tackling the MBEL problem with a discriminative classifier, we formulate it as a sequence-to-sequence generation task, which better exploits the shared dependencies between source mentions and target entities. Moreover, Con2GEN matches against UMLS concepts in as many languages and types as possible, hence facilitating cross-information disambiguation. Extensive experiments show that our model achieves promising performance improvements compared with several state-of-the-art techniques on the XL-BEL and the Mantra GSC datasets spanning 12 typologically diverse languages.

## 1 Introduction

Multilingual biomedical entity linking (MBEL) is an essential yet challenging task for domain-specific natural language understanding, which refers to the process of mapping mentions in a source language to standard entities in a multilingual knowledge base (KB). Successes in building MBEL systems can assist various medical research, such as biomedical knowledge probing (Meng et al., 2022) and clinical decision support (Chen et al., 2022), which need to determine the distinct meanings of biomedical terms, regardless of language.

Although promising breakthroughs have been achieved in monolingual biomedical entity linking

---
*Corresponding authors.

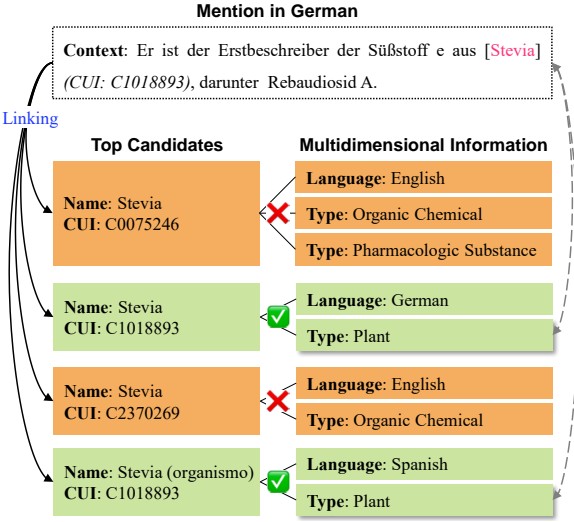

Figure 1: An example of multilingual biomedical entity linking, where the mention is in red. The colored boxes represent the top candidates retrieved by SAPBERT$_{multi}$. Among them, the orange box indicates the incorrectly linked entity and the green box indicates the matching entity.

(typically English) (Bitton et al., 2020; Liu et al., 2021a; Zhu et al., 2022), these approaches cannot be effectively applied to other languages due to the huge discrepancies between multilingual and monolingual versions of the biomedical entity linking (BEL) task. Firstly, resources for non-English BEL are scarce, hindering the development of multilingual research. For example, although the Unified Medical Language System (UMLS) (Bodenreider, 2004) is often regarded as a large multilingual KB in the biomedical domain, more than 70% of concept synonyms are in English, while the minority of concept names are in other languages. Secondly, the *ambiguity* problem in MBEL is more severe compared to monolingual BEL, since named entities with the same surface form in different languages can express distinct semantics, which makes the task of MBEL more challenging.

A dominant paradigm of most current MBEL

solutions is to provide a single representation for each biomedical term by fine-tuning multilingual pre-trained models, such as mBERT (Devlin et al., 2019) or XLM-R (Conneau et al., 2020), on phrase-level synonyms extracted from the UMLS (Liu et al., 2021b; Yuan et al., 2022b). However, these methods still suffer from two major drawbacks, namely *indirect* and *uninformative*. Concretely, the *indirect* problem refers to the candidate entities are ranked with a dot-product between dense vectors between the mention and the entity, which fails to capture the interactions between them. The *uninformative* problem refers to the fact that current work solves the MBEL task via shallow matching based on surface forms, which ignores the deep underlying information of entities, such as language and context.

Consider the example in Figure 1, where *Stevia* is a German mention of the concept *C1018893* in UMLS. In this case, although the top 1 candidate entity retrieved by SAPBERT$_{multi}$ (Liu et al., 2021b) has the same surface form as the mention, it is not the target entity. Specifically, the entity *Stevia* is an English synonym for the concept *C0075246*, of type *Organic Chemical* or *Pharmacologic Substance*, which is an ambiguous instance across languages and types. However, if the MBEL model understands the context of mentions and utilizes the language and type information of entities, the target concept may have a greater chance of being retrieved, which will make the entity disambiguation across languages more precise.

Motivated by the above observations, in this paper, we propose a prompt-based controllable contrastive generation (Con2GEN) algorithm to disambiguate biomedical entities across diverse languages. To address the indirect challenge, we exploit a sequence-to-sequence architecture to effectively cross-encode the mention and the entity to capture more latent interactions. Generative models inherently have a better ability to align input and output spaces than discriminative models, since discriminative models need to iteratively fit atomic labels to learn aligned features (Cao et al., 2022). In detail, Con2GEN summarizes multidimensional information of the UMLS concept mentioned in biomedical text into a natural sentence following a predefined template, mainly aiming to solve the uninformative challenge and reduce ambiguity. Specifically, the predefined template is as close as possible to natural language and

matches against UMLS concepts in as many languages and types as possible, which allows the transfer of the latent knowledge of multilingual pre-trained models about the task. In this way, the decoder of Con2GEN can be guided to generate in a limited space by filling the fixed slots in the predefined template. Moreover, we utilize contrastive learning to alleviate the exposure bias problem in generation (An et al., 2022) — an autoregressive model trained only using the ground truths exhibits inferior generalization performance. Finally, we evaluate the performance of our model using the XL-BEL (Liu et al., 2021b) and the Mantra GSC (Kors et al., 2015) datasets spanning 12 typologically diverse languages, and against several state-of-the-art (SOTA) baselines. From the extensive experimental results, the proposed Con2GEN strategy is demonstrated to be effective in multilingual disambiguation based on multidimensional information of bio-entities. The code is available at https://github.com/TiantianZhu110/Con2GEN.

To sum up, the contributions of this paper are as follows: (1) For the first time, we propose the Con2GEN, a novel controllable contrastive generative method, to address the ambiguity challenge of the MBEL task; (2) We design a constrained decoding algorithm consisting of a predefined template and a contrastive learning mechanism for entity multidimensional information injection during inference; (3) Extensive experiments show that our model achieves promising performance improvements on three public MBEL datasets spanning 12 diverse languages.

## 2 Related Work

### 2.1 Multilingual Biomedical Entity Linking

Multilingual biomedical entity linking (MBEL) task is an extended version of monolingual biomedical entity linking (Zhu et al., 2022; Liu et al., 2021a; Wu et al., 2020) and cross-lingual biomedical entity linking (XBEL) (Rijhwani et al., 2019; Bitton et al., 2020), which focuses on mapping an input mention from biomedical text in a specific language to its associated entity in a curated multilingual KB. In monolingual biomedical entity linking, mentions always match the KB language, and entities in other languages are discarded. However, the monolingual formulation of biomedical entity linking has the problem that the KB might miss several entries for languages with limited coverage of entity descriptors. The XBEL formulation

alleviates this problem to some extent and considers mentions expressed in different languages and linked to a monolingual KB (usually English) (Roller et al., 2018; Bitton et al., 2020). Although both the MBEL and XBEL formulations exploit inter-language links to identify entities in other languages, XBEL requires the target KB to be monolingual, which might still miss several entries in the KB.

In this work, we focus on MBEL, which is more general since it can link to entities that might not be necessarily represented in the target monolingual KB but in any of the available languages. In particular, we use UMLS (Bodenreider, 2004) as our multilingual biomedical KB, as it is a rich conceptual ontology differentiated by Concept Unique IDs (CUIs) and relations between them across 25 languages. Moreover, we maintain as much language and type information as possible, hence learning the connections between the source language and entity names in different languages.

## 2.2 Sequence to Sequence Text Generation

The sequence-to-sequence models (Sutskever et al., 2014) learn to generate an output sequence of arbitrary length from an input sequence, and have demonstrated its flexibility and effectiveness in a wide range of tasks compared to classification-based methods. Early generative models used recurrent neural networks (RNNs) and attention mechanism (Bahdanau et al., 2015) to capture long-term dependencies between the source and the target sequences. Later, the Transformer (Vaswani et al., 2017) was proposed, which is solely based on attention mechanisms, dispensing with recurrence and convolutions, and showed significant performance improvement. Recently, Transformer based pre-trained language models (De Cao et al., 2021; Lewis et al., 2020; Raffel et al., 2020; Liu et al., 2020) provide a new trend to utilize massive text corpora and have become the baseline for most text generation tasks due to their impressive performance.

Close to this work, mGENRE (Cao et al., 2022) also employed neural generation models for multilingual entity linking, mapping mentions to the Wikipedia titles in multiple languages. However, mGENRE focuses on multilingual entity linking in the general domain and cannot be simply transferred to the biomedical domain. Furthermore, mGENRE applies a sequence-to-structure decoding algorithm whose output sequences contain non-semantic indicators such as ">>". However, such non-semantic indicators express little semantic information and have a gap with natural language, which may mislead the decoding process. Unlike the decoding algorithm of mGENRE, Con2GEN generates the associated entities by filling the fixed slots in the predefined natural language template, which allows Con2GEN to handle various entity types and alleviate ambiguity in biomedical tasks.

Furthermore, GenBioEL (Yuan et al., 2022a) is the first to explore generative EL in the biomedical domain. However, the differences between our work and GenBioEL are as follows: (1) Different decoding objectives: GenBioEL only generates the entity name at the decoding phase, ignoring the multidimensional information of bio-entities and increasing the difficulty of disambiguation. Instead, we proposed a constrained decoding algorithm consisting of a predefined template for multidimensional information injection during inference. (2) Different training manners: GenBioEL pre-trained the model to improve the generalization ability. Instead, we utilized multi-task learning to jointly optimize the contrastive loss and the task loss and meanwhile to strengthen the proposed Con2GEN model.

In this paper, we aim to facilitate MBEL with a sequence-to-sequence generation method to capture the dependencies between mentions and entities. Specifically, we propose a novel prompt-based controllable contrastive decoding algorithm, which can guide the generation process using a predefined natural language template and contrastive learning. In this way, the multidimensional knowledge of biomedical entities can be injected and exploited during inference.

## 3 Method

### 3.1 Problem Statement

Given a biomedical mention $m$ in free text $x$ of language $l$, and a multilingual KB (e.g., UMLS) of $n$ entities $E = \{e_1, e_2, \ldots, e_n\}$, the MBEL system will output the target entity $e_t$ from $E$. In this work, we assume that each mention has its own golden CUI and each CUI has at least one synonym, regardless of language. Note that the type and language information of the bio-entities is required, which differs from the general domain setting. In the general domain, we assume that each entity descriptor contains a name that concisely describes the entity,

e.g., the Wikipedia title (with language identifier) can be used to uniquely identify an entity descriptor, while in the biomedical domain entity names with languages are still ambiguous combinations. Note that mapping the output to CUI is a non-trivial task. However, by adding multidimensional information, ambiguity has been greatly reduced. In the experiment, if the combination of entity name, type, and language cannot uniquely identify a CUI, we will randomly select one CUI from them. More details are provided in Appendix A.1.

## 3.2 Model Architecture

In this work, we frame the MBEL task as a sequence-to-sequence generation problem and the overall architecture is illustrated in Figure 2. Given the token sequence $x = x_1, \ldots, m, \ldots, x_{|x|}$ of the mention along with contextual information, Con2GEN directly generates the most relevant entity $e_t$ according to a specific template $T(\cdot)$ via an encoder-decoder architecture. Concretely, Con2GEN first encodes the input $x$, and then uses a prompt-based controllable contrastive decoding algorithm to generate the corresponding entity in natural language: a sequence of tokens $y$. In the following sections, we introduce in detail the neural conditional text generation network and the prompt-based controllable contrastive decoding algorithm proposed in this work.

## 3.3 Neural Conditional Text Generation Network

A neural conditional text generation model (Sutskever et al., 2014) $\mathcal{M} = (f, g)$ generates the target sequence $y$ with respect to the source sequence $x$, where $f$ and $g$ denotes the encoder and decoder, respectively. To this end, Con2GEN first encodes the input biomedical sequence $x$ into the hidden vector representations $H^e = h_1^e, \ldots, h_{|x|}^e$ via a multi-layer transformer encoder $f$:

$$H^e = f(x_1, \ldots, x_{|x|}) \tag{1}$$

During decoding, $g$ utilizes the sequential hidden vectors $H^e$ to predict the target entity token-by-token. Specifically, at step $i$, the token $y_i$ and the decoder state $h_i^d$ are computed as follows:

$$y_i, h_i^d = g([H^e; h_1^d, \ldots, h_{i-1}^d]; y_{i-1}) \tag{2}$$

Therefore, the entire conditional probability $p(y|x)$ of generating the output sequence $y$ given the input sequence $x$ can be progressively calculated as

follows:

$$P_\theta(y|x) = \prod_i^{|y|} p_\Theta(y_i|y_{<i}, x) \tag{3}$$

where $y_{<i} = y_1 \ldots y_{i-1}$, $y$ is the identifier of entity, and $\Theta$ denotes the parameters of the model $\mathcal{M}$. Moreover, since the input and the output tokens of our model are both natural language words, we adopt the multilingual pre-trained language model mBART (Liu et al., 2020) as our base model, which will take advantage of the general multilingual text generation knowledge.

## 3.4 Prompt-based Controllable Decoding

### 3.4.1 Template

By designing an appropriate template, we encourage Con2GEN to better utilize the multidimensional information of entities, therefore, to solve the uninformative problem. The desired prompt not only provides information but also defines the output format. Specifically, we create a natural language template $T(\cdot)$, in which we list all meaningful elements of an entity in the format of placeholders: $T =$ [Ent] of type [Ent Type] in [Ent Lang]. Among them, [Ent] denotes the entity name. [Ent Type] is the semantic type of the entity and we use an empty string for entities without type information in UMLS. [Ent Lang] denotes the target language.

### 3.4.2 Controllable Decoding

Naturally, the most straightforward solution for decoding is the greedy algorithm (Germann et al., 2001), which selects the token with the highest probability $p$ at each step $i$. However, the greedy decoding algorithm may cause invalid entities that are not in the vocabulary of the UMLS. In addition, the greedy decoding algorithm is uncontrollable since it cannot output sequences in the template format we designed, further ignoring the useful knowledge of entities.

Taking the above needs into consideration, we employ a controllable decoding algorithm based on the trie tree (Cormen et al., 2009; Lu et al., 2021) to only output the valid token from a constrained vocabulary set at each decoding step $i$. During controllable decoding, the multidimensional information of entities is injected into the predefined template as a prompt for the decoder. Specifically, at each step $i$, the trie-based decoding algorithm will dynamically traverse the children nodes from

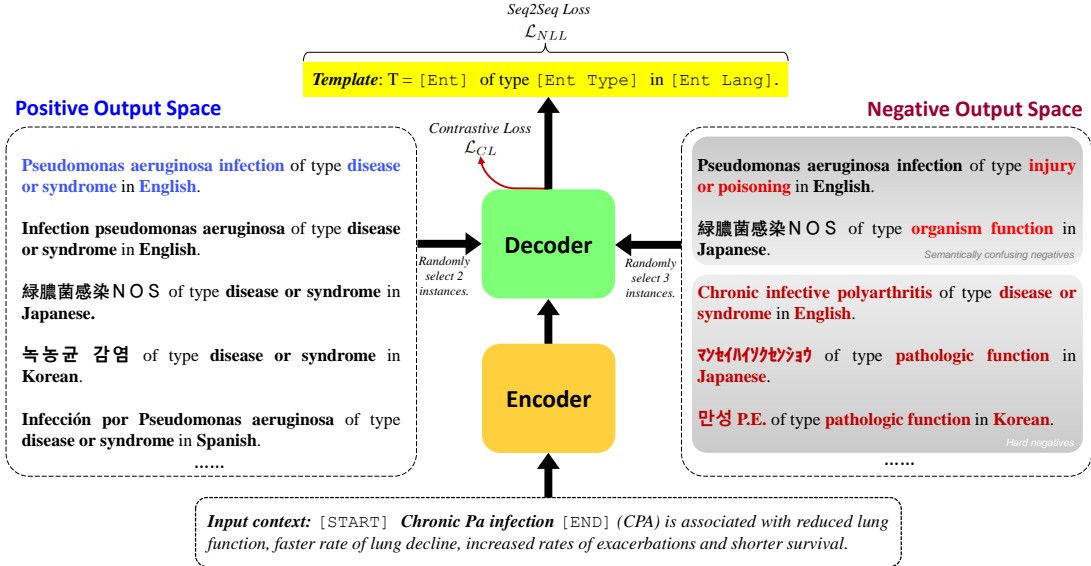

Figure 2: Con2GEN: the input is a biomedical text where the mention is signaled with special tokens [START] and [END], and the output is the entity identifier following the predefined template. We use a prompt-based controllable decoder to generate entity names as well as types and languages. This example is a real instance from our system.

the current state. For ease of understanding, we give an example of a complete decoding process by executing a trie tree search in Appendix A.2.

Since the vocabulary set of UMLS $E$ is very large (over 10M deduplicated entities) and the cost of ranking each entity in $E$ is prohibitive, we further restrict the generation space at inference time. First, we employ the beam search approximate strategy (Sutskever et al., 2014) to search for the top-$b$ entities in $E$ instead of scoring all entities, which greatly improves inference efficiency. Moreover, modern entity linking systems (Ledell Wu, 2020; Zhu et al., 2022) tend to apply candidate retrieval first and then apply a finer-grained reranking instead of scoring all entities at once in order to save computational cost. Inspired by this, we also leverage the SAPBERT$_{multi}$ (Liu et al., 2021b) to generate top-$k$ candidates with similar surface form for each input mention and further constrain the decoding generation space.

### 3.5 Training

### 3.5.1 Contrastive Learning

Typically, generation systems are trained with teacher forcing with the ground truth labels in the training phase and are exposed to incorrectly generated tokens during the inference phase, thus encountering the exposure bias problem (An et al., 2022). Although the traditional neural generation method no longer needs negative sampling, the trained model will lose the ability to discriminate

hard negative samples during inference. Recently, contrastive learning provides a new idea for alleviating the exposure bias problem of generation models by additionally constructing positive and negative pairs during training (An et al., 2022). Contrastive learning optimizes model parameters by learning an efficient representation space for the data, in which positive pairs are pulled together and negative pairs are pushed apart.

In this paper, we expect that by contrasting hard

---

**Algorithm 1** Contrastive learning training strategy

**Input**: Initialize the generation model $\mathcal{M}$ as: $\Theta' = \Theta$, one training instance as $d = (x, y)$, where $x$ is a input sequence, $y$ is an output sequence

1: **for** epoch $e = 1$ to $E$ **do**
2:     **for** batch $b = 1$ to $B$ **do**
3:        Batch data $D^b = \{d_1^b, d_2^b, ..., d_{|D^b|}^b\}$
4:        **for** $d_j^b \in D^b$ **do**
5:           Randomly select two positive outputs $y_j^p$, $y_j^{p1}$ from the synonym set of the same CUI
6:           Randomly select three negative outputs $y_j^{n1}$, $y_j^{n2}$, $y_j^{n3}$ from the union of hard negatives and samples with type information replaced
7:           Compute representations $h_j^p = \Theta(x_j, y_j^p)$, $h_j^{p1} = \Theta(x_j, y_j^{p1})$, $h_j^{n1} = \Theta(x_j, y_j^{n1})$, $h_j^{n2} = \Theta(x_j, y_j^{n2})$, $h_j^{n3} = \Theta(x_j, y_j^{n3})$
8:           Compute contrastive loss $\mathcal{L}_{CL}$ with $h_j^p$, $h_j^{p1}$, $h_j^{n1}$, $h_j^{n2}$, $h_j^{n3}$
9:        **end for**
10:       Compute average batch loss and update the parameter $\Theta$
11:     **end for**
12: **end for**

negatives, the proposed Con2GEN model can be sensitive to the entity semantics and further alleviate the exposure bias problem. Towards this, we first create the positive and negative pairs for each training sample. Specifically, for a specific training sample $(x, y)$, we keep the input $x$ constant and vary the output $y$. Then, we filter out the synonyms in different languages with the same CUI as $y$ to build the positive set. Moreover, the negative instances come from two sources: 1) we replace the type information in the template of $y$ in order to create the semantically confusing negative instances; 2) we regard the incorrect candidates retrieved by SAPBERT$_{multi}$ as hard negative instances.

In the training phase, for each training sample $(x, y)$, we randomly select two positive samples $(x, y^p)$, $(x, y^{p_1})$ and three negative samples $(x, y^{n_1})$, $(x, y^{n_2})$, $(x, y^{n_3})$ from the additionally constructed positive and negative sets, respectively. Then we adopt the average outputs of $\mathcal{M}$ as the instance representation for modeling the contrastive objective. Inspired by InfoNCE (van den Oord et al., 2018), we define an objective $\mathcal{L}_{CL}$ in the contrastive manner:

$$\mathcal{L}_{CL} = -log \frac{e^{sim(h^p, h^{p_1})}}{e^{sim(h^p, h^{p_1})} + \sum_{i=1}^{3} e^{sim(h^p, h^{n_i})}} \quad (4)$$

where $h^p = \Theta(x, y^p)$, $h^{p_1} = \Theta(x, y^{p_1})$, $h^{n_1} = \Theta(x, y^{n_1})$, $h^{n_2} = \Theta(x, y^{n_2})$, $h^{n_3} = \Theta(x, y^{n_3})$, $\Theta$ denotes the parameters of the model $\mathcal{M}$. The details of the contrastive learning training process are described in Algorithm 1.

### 3.5.2 Training Objective

This section describes the training objective of the proposed Con2GEN model. Specifically, given a training instance $(x, y)$ pair, where $x$ is the source sequence, and $y$ is the target sequence, we first compute the following negative log-likelihood (NLL) loss:

$$\mathcal{L}_{NLL} = -\sum_{i=1}^{|y|} \log p_\theta(y_i | y_{<i}, x) \quad (5)$$

where $\theta$ is the parameters of model $\mathcal{M}$. Simultaneously, for each instance, we compute the contrastive loss $\mathcal{L}_{CL}$ as described in the previous section. Hence, the final loss function $\mathcal{L}$ is the sum of the two, where $\mathcal{L} = \mathcal{L}_{NLL} + \mathcal{L}_{CL}$. By combining the NLL and contrastive loss in the training phase, the Con2GEN model can be jointly optimized in a multi-task learning manner, which will improve the generalization ability of the model. Moreover,

we explored different ways to combine the NLL and contrastive loss. We utilized a hyper-parameter "label weight" to balance the NLL and contrastive loss, and did a grid search from 0 to 1 with a step size of 0.1. And we found when the label weight equals 0.5, the performance is best on the validation set.

## 4 Experiments

### 4.1 Datasets

We used three datasets to evaluate the proposed Con2GEN model. Descriptions of the datasets and their statistics are provided in Table 1. **XL-BEL** (Liu et al., 2021b) contains sentences from 10 language dumps of Wikipedia, with mentions linked to the 2020AA full version of UMLS concepts. The size of the concept set in UMLS is $4,281,184$, and the size of all surface form/synonym set is $12,671,936$. And the concepts in UMLS belong to 127 semantic types. For each language, $1,000$ instances are randomly selected for the final test sets, where each evaluation instance consists of a triplet $(sentence, mention, CUI)$. **EMEA** and **Patent** are two subsets of the Multilingual Annotation of Named Entities and Terminological Resource Acquisition gold-standard corpora (the Mantra GSC) (Kors et al., 2015), which belong to different text types: European Medicines Agency (EMEA) and Patent. The Mantra GSC contains French, German, Spanish, and Dutch corpora that include medical terms mapped to standard terms in SNOMED-CT, MeSH, and MedDRA of the 2020AA full version of UMLS, including $801,038$ CUIs.

Moreover, due to the lack of training data for the MBEL task, we utilize the Medmentions English biomedical entity linking dataset (Mohan and Li, 2019) to construct the multilingual contextual training data. In order to solve the problem that the input of training data is all in English, we used the weakly supervised data augmentation method to increase the multilingual characteristics of the training data. Specifically, we swap the English mention of input with the multilingual entity of output. Since the Medmentions dataset only contains English instances, we use this simple method to generate multilingual input. We preserve as many multilingual entities as possible, which plays a role in smoothing multilingual data. We didn't use any translator in this process, which will not introduce any extra noise. The final constructed training and validation sets contain $3,136,568$ and $412,905$ in-

| Dataset→ | Train | Dev | XL-BEL | | | | | | | | | | EMEA | | | | Patent | |
|---|---|---|---|---|---|---|---|---|---|---|---|---|---|---|---|---|---|---|
| Num.↓ | | | EN | ES | DE | FI | RU | TR | KO | ZH | JA | TH | ES | FR | NL | DE | FR | DE |
| Sentences | 30,349 | 10,058 | 994 | 986 | 996 | 995 | 995 | 966 | 971 | 988 | 984 | 963 | 100 | 100 | 100 | 100 | 50 | 50 |
| Mentions | 3,136,568 | 412,905 | 1000 | 1000 | 1000 | 1000 | 1000 | 1000 | 1000 | 1000 | 1000 | 1000 | 432 | 431 | 434 | 425 | 342 | 348 |
| Entities | 25,529 | 12,535 | 807 | 889 | 908 | 816 | 816 | 694 | 829 | 875 | 822 | 704 | 295 | 304 | 304 | 299 | 223 | 223 |

Table 1: Statistics of different multilingual biomedical entity linking datasets used in this work.

stances, respectively.

## 4.2 Evaluation Metrics and Baselines

For evaluation, we report Accuracy for evaluating the performance of the proposed model against the baseline models following the previous work (Cao et al., 2022). Appendix A.3 gives detailed descriptions of baselines.

## 4.3 Implementation Details

We used Adam optimizer with a learning rate of $1e - 6$, $\beta_1 = 0.9$, $\beta_2 = 0.999$, eps$= 1e - 8$, and with warm up for 500 steps for model training. The dropout rate and attention dropout are set to 0.1 and 0.1. We clip the gradient norm within 0.1. The label smoothing of NLL loss is 0.1 and weight decay is 0.01. We used max 1,024 tokens and variable batch size($\approx 15$). The training epoch of the model are set to 3. At test time, we use Constrained Beam Search with 7 beams. We implemented our model using the fairseq library (Ott et al., 2019). Note that all parameters are selected based on the best performance on the development set.

## 4.4 Experimental Results

### 4.4.1 Main Results

To evaluate our approach, we compared the proposed Con2GEN model with several SOTA models on the XL-BEL and the Mantra GSC EMEA/Patent datasets and listed the performance of each model in Table 2. As shown in Table 2, our model outperformed all other models, which improved 1.3 Accuracy points on average over the baselines on XL-BEL, EMEA and Patent test sets. We note that the discriminative PLMs (mBERT, XLM-R) underperformed the generative PLM (mBART), which demonstrates the advantage of generative models in solving the indirect problem of MBEL. Since Bio-PRO (Zhu et al., 2022) is a context-infused method in the monolingual BEL task, we finetuned it on the multilingual training data and reported the results in Table 2. Note that mGENRE (Cao et al., 2022) is a SOTA method in the general domain, which

was not compared in most biomedical studies. We compared with both the base and finetuned results of mGENRE in Table 2.

In addition, to analyze the contribution of each part of our model, we cumulatively removed components and listed the performance comparison of the proposed model and its variants in Table 2. The w/o type, w/o lang, and w/o type & lang methods represent the use of partial dimensional information of entities rather than multidimensional information, resulting in a drop on performance. Even so, Con2GEN's variants with partial dimensional information are still strong baselines compared with other baselines. The contrastive mechanism is removed from the w/o CL method, which drops the Accuracy point of test sets compared with Con2GEN. The contrastive learning strategy can identify the hard examples and is demonstrated to be effective on 12 language subsets from Table 2. We believe the improvements are significant since we only trained one single model which outperforms baselines on almost all languages. Although the contrastive learning strategy improved the average accuracy by only 0.2 points, it worked across 9 diverse languages, which is not trivial. Moreover, we added an ablation experiment of Con2GEN without using the controllable decoding algorithm, and the performance dropped significantly, which demonstrates the effectiveness of the proposed controllable decoding. In conclusion, the template with both language and type information is better than other templates in generation and the model that integrates contrastive learning and controllable decoding has the best performance.

### 4.4.2 Resolution of the Ambiguity Problem

In order to study the resolution of different models on the ambiguity problem, we compared the top 1 candidates prediction's performance of our proposed model with $\text{SAPBERT}_{multi}$, which is a surface form matching based MBEL model. As mentioned before, the ambiguous phenomenon is severe in the MBEL task, which exists not only in different contexts but also across languages, in-

| Dataset→ | XL-BEL | | | | | | | | | | EMEA | | | | Patent | | Avg |
|---|---|---|---|---|---|---|---|---|---|---|---|---|---|---|---|---|---|
| Method↓ | EN | ES | DE | FI | RU | TR | KO | ZH | JA | TH | ES | FR | NL | DE | FR | DE | |
| mBERT (Devlin et al., 2019) | 29.1 | 11.9 | 5.2 | 1.6 | 5.1 | 8.0 | 0.6 | 3.6 | 9.1 | 0.9 | 4.4 | 10.0 | 4.8 | 4.0 | 7.0 | 4.6 | 6.9 |
| XLM-R (Conneau et al., 2020) | 0.5 | 0.0 | 0.0 | 0.0 | 0.0 | 0.0 | 0.0 | 0.0 | 0.0 | 0.0 | 0.0 | 0.0 | 0.0 | 0.0 | 0.0 | 0.0 | 0.0 |
| mBART (Liu et al., 2020) | 34.8 | 23.5 | 15.0 | 11.7 | 20.4 | 24.9 | 9.0 | 8.3 | 13.1 | 11.2 | 35.6 | 30.9 | 29.0 | 29.9 | 27.2 | 29.0 | 22.1 |
| BioPRO+finetune (Zhu et al., 2022) | 80.4 | 46.0 | 27.2 | 19.0 | 34.2 | 34.3 | 12.7 | 12.7 | 19.8 | 12.9 | 53.9 | 50.3 | 47.5 | 46.8 | 55.0 | 50.0 | 37.7 |
| SAPBERT$_{multi}$ (Liu et al., 2021b) | 86.2 | 48.4 | 28.5 | 19.9 | 38.4 | 35.5 | 14.8 | 14.5 | 23.1 | 14.0 | 58.3 | 53.4 | 55.5 | 53.6 | 63.5 | 61.2 | 41.8 |
| CODER (Yuan et al., 2022b) | 85.9 | 51.4 | 31.0 | 20.3 | 37.3 | 35.4 | 1.4 | 14.4 | 22.6 | 2.4 | 56.9 | 53.6 | **56.9** | 53.4 | 64.9 | 65.8 | 40.9 |
| mGENRE (Cao et al., 2022) | 85.2 | 48.9 | 30.9 | 23.8 | 39.9 | 36.9 | 15.7 | 16.7 | 23.9 | 16.0 | 56.5 | 54.8 | 49.3 | 52.7 | 61.1 | 58.6 | 41.9 |
| mGENRE+finetune (Cao et al., 2022) | 85.4 | 49.8 | **31.3** | 24.8 | **41.5** | 39.5 | 16.0 | 16.9 | 23.9 | 17.2 | 62.0 | 56.4 | 55.1 | 54.1 | 61.2 | 55.7 | 43.2 |
| Con2GEN (w/o type) | 84.3 | 51.3 | 30.4 | 24.6 | 38.0 | 39.8 | 16.9 | **17.6** | 24.2 | 18.7 | 59.5 | 54.5 | 49.5 | 52.5 | **66.4** | 65.5 | 43.4 |
| Con2GEN (w/o lang) | 86.0 | 52.1 | 31.2 | 24.7 | 39.5 | 40.4 | 17.1 | 17.3 | 25.3 | 18.2 | 63.2 | **57.8** | 54.1 | 55.1 | 64.9 | 64.7 | 44.5 |
| Con2GEN (w/o type & lang) | 82.9 | 48.6 | 30.3 | 24.0 | 37.8 | 39.5 | 16.8 | 16.6 | 24.5 | 18.0 | 56.5 | 55.5 | 52.1 | 53.4 | 66.4 | 65.5 | 43.0 |
| Con2GEN (w/o CL) | 86.3 | 52.2 | 30.6 | **24.9** | 39.7 | 40.3 | 17.2 | 17.3 | 25.4 | 18.9 | 63.0 | 55.9 | 51.8 | 54.8 | 64.9 | 64.9 | 44.3 |
| Con2GEN (w/o controllable decoding) | 52.7 | 4.5 | 5.0 | 2.0 | 3.3 | 7.1 | 5.3 | 6.4 | 12.5 | 4.6 | 13.2 | 20.0 | 17.1 | 15.1 | 23.4 | 20.7 | 13.3 |
| Con2GEN | **86.6** | **52.7** | 30.8 | 24.4 | 39.8 | **40.5** | **17.4** | 17.1 | **25.6** | **19.2** | **63.9** | 57.1 | 51.4 | **55.5** | 64.0 | **66.1** | **44.5** |

Table 2: Performance on the XL-BEL and the Mantra GSC EMEA / Patent datasets in comparison with the SOTA methods. Avg refers to the average performance across all target languages. Top: SOTA methods. Bottom: Con2GEN with and without different components. "w/o type" denotes removing type information from the template. "w/o lang" denotes removing language information from the template. "w/o type & lang" denotes removing type and language information from the template. "w/o CL" denotes training without the contrastive learning strategy. "w/o controllable decoding" denotes without using the controllable decoding. Bold denotes the best result in the column.

| Dataset→ | XL-BEL | | | | | | | | | | Avg |
|---|---|---|---|---|---|---|---|---|---|---|---|
| Method↓ | EN | ES | DE | FI | RU | TR | KO | ZH | JA | TH | |
| mGENRE | 85.2 | 48.9 | 30.9 | 23.8 | 39.9 | 36.9 | 15.7 | 16.7 | 23.9 | 16.0 | 33.8 |
| +finetune | 85.4 | 49.8 | 31.3 | 24.8 | 41.5 | 39.5 | 16.0 | 16.9 | 23.9 | 17.2 | 34.6 |
| +finetune+type | 87.7$^{\uparrow}$ | 50.7$^{\uparrow}$ | 30.4 | 24.5 | 41.2 | 40.1$^{\uparrow}$ | 16.7$^{\uparrow}$ | 16.4 | 25.3$^{\uparrow}$ | 18.0$^{\uparrow}$ | 35.1$^{\uparrow}$ |
| +finetune+CL | 85.3 | 50.0$^{\uparrow}$ | 31.7$^{\uparrow}$ | 24.8 | 42.5$^{\uparrow}$ | 39.9$^{\uparrow}$ | 16.4$^{\uparrow}$ | 16.6 | 24.3$^{\uparrow}$ | 17.5$^{\uparrow}$ | 34.9$^{\uparrow}$ |
| +finetune+type+CL | 86.8$^{\uparrow}$ | 50.9$^{\uparrow}$ | 30.7 | 24.8 | 41.9$^{\uparrow}$ | 40.2$^{\uparrow}$ | 16.9$^{\uparrow}$ | 16.7 | 24.9$^{\uparrow}$ | 18.3$^{\uparrow}$ | 35.2$^{\uparrow}$ |

Table 3: Accuracy of mGENRE's variants on the XL-BEL dataset. "+finetune" denotes finetuning on the multilingual biomedical training data we constructed. "+finetune+type" denotes finetuning with type information. "+finetune+CL" denotes finetuning with the contrastive learning strategy. "+finetune+type+CL" denotes finetuning with type information and contrastive learning strategy.

creasing the challenge of the task. Table 4 lists the number of wrongly predicted top 1 candidates, the number of ambiguous examples in wrongly predicted top 1 candidates, and the proportion of ambiguous instances in the overall top 1 errors on the XL-BEL test set. It is shown that our model can greatly reduce the proportion of ambiguous examples among the total misclassified examples, which demonstrates the effectiveness of the proposed method in addressing the ambiguity challenge and understanding the deep semantics of mentions and entities.

### 4.4.3 Effect of Multidimensional Information Injection

From Table 2, we observe that mGENRE is a strong baseline even without finetuning on the multilingual biomedical dataset. To further study the potential of mGENRE on the MBEL task, we finetuned it on the training data we constructed. As can be seen from Table 3, "+finetune" method outper-

forms the base mGENRE model on each XL-BEL test set, which demonstrates the effectiveness of the proposed weakly supervised data augmentation method. Moreover, we assume that the injection of multidimensional information may be useful to the mGENRE generation as well. Thus, we reported the "+finetune+type" results in Table 3, which improves the accuracy by 0.5 on average compared to the base model. Note that the "+finetune+type" variant of mGENRE still underperformed our model, which demonstrates the effectiveness of the proposed natural language template, since both mGENRE and our Con2GEN utilize controllable decoding at inference, yet their output templates differ.

### 4.4.4 Effect of Contrastive Learning

As shown in Table 2, the proposed model achieved better performance with the contrastive learning strategy compared with the "w/o CL" method on 12 subsets. To further study the effect of contrastive

| Source | Mention with context | SAPBERT$_{multi}$ | Con2GEN |
|---|---|---|---|
| DE | In den Gattungen "Legionella", "Pseudomonas", "Vibrio", "Salmonella" und "[START] **Shigella** [END]" findet man wichtige Krankheitserreger von Mensch, Tier und Pflanze. | Shigella \| EN \| Disease or syndrome | Shigella \| DE \| Bacterium |
| ES | La Fórmula de Parkland es una fórmula utilizada para estimar la cantidad de fluido de reposición requerida para las primeras 24 horas en un paciente [START] **quemado** [END] a fin de garantizar que permanezca hemodinámicamente estable. | Quemante \| ES \| Qualitative concept | Quemadura (trastorno) \| ES \| Injury or poisoning |
| FI | Esimerkiksi virginianopossumeja on viety Pohjois-Amerikan länsiosiin ja Uuteen-Seelantiin , punakaulakenguru ita Britanniaan sekä [START] **parmavallebeja** [END] , damavallabeja ja kettukusu ja Uuteen-Seelantiin. | Parvamoeba \| EN \| Eukaryote | Macropus parma (organismo) \| ES \| Mammal |
| JA | 母の実家は [START] **クインシーメロン** [END] の農家であり、そのため正代の実家の冷蔵庫には商品にならないメロンが大量にあった。 | クインスロン \| JA \| Amino acid, peptide, or protein | メロン \| JA \| Plant |
| KO | 당밀 성분을 함유하고 있으며, 자당 50%, 전화당 ( 포도당 과 과당 ) 20%, [START] **수분** [END] 20%, 그리고 회분, 단백질, 버개스 섬유질 등으로 이루어져 있다. | 수분 \| KO \| Body location or region | Producto que contiene agua (producto medicinal) \| ES \| Pharmacologic substance |
| RU | Доказал целесообразность применения костно-хрящевых алло- и [START] **ксенотрансплантантов** [END] . | Ксенотрансплантаты \| RU \| Biomedical or dental material | Ксенотрансплантация \| RU \| Therapeutic or preventive procedure |
| TR | Kromat işçilerinde akciğer [START] **malignantlarının** [END] en sık görülen şekli, skuamöz hücreli karsinomdur. | Malignant \| EN \| Qualitative concept | 悪性腫瘍 \| JA \| neoplastic process |
| ZH | 其与主流粑粑的区别是，官渡粑粑是用芝麻、花生 、[START] **核桃仁** [END] 磨碎作馅，用白糖或者红糖烘烤融化。 | 核仁 \| JA \| Cell component | 胡桃 \| JA \| Plant |

Figure 3: Examples of top 1 candidates retrieved by SAPBERT$_{multi}$ and Con2GEN. Bold denotes mentions. Underline refers to the golden entities.

| Method→ | SAPBERT$_{multi}$ | | | Con2GEN | | |
|---|---|---|---|---|---|---|
| Language↓ | #Err | #Amb | Percent | #Err | #Amb | Percent |
| EN | 138 | 87 | 63.0% | 134 | 56 | 41.8%$^{\downarrow}$ |
| ES | 516 | 108 | 20.9% | 473 | 37 | 7.8%$^{\downarrow}$ |
| DE | 715 | 4 | 0.6% | 692 | 2 | 0.3%$^{\downarrow}$ |
| FI | 801 | 12 | 1.5% | 756 | 4 | 0.5%$^{\downarrow}$ |
| RU | 616 | 21 | 3.4% | 602 | 5 | 0.8%$^{\downarrow}$ |
| TR | 644 | 16 | 2.5% | 595 | 8 | 1.3%$^{\downarrow}$ |
| KO | 852 | 42 | 4.9% | 826 | 30 | 3.6%$^{\downarrow}$ |
| ZH | 855 | 24 | 2.8% | 829 | 17 | 2.1%$^{\downarrow}$ |
| JA | 769 | 152 | 19.8% | 744 | 102 | 13.7%$^{\downarrow}$ |
| TH | 859 | 2 | 0.2% | 808 | 1 | 0.1%$^{\downarrow}$ |
| Avg | 676.5 | 46.8 | 6.9% | 645.9 | 26.2 | 4.1%$^{\downarrow}$ |

Table 4: Statistics of the ambiguous instances in mispredictions on XL-BEL test set. "#Err" refers to the number of top 1 candidates predicted incorrectly. "#Amb" refers to the number of ambiguous instances in top 1 candidates predicted incorrectly. "Percent" refers to the ratio of ambiguous instances to error instances.

learning, we had initially hoped that the contrastive learning's capability would make other neural networks work better, and we add it to the mGENRE during finetuning. In Table 3, the improvements of the mGENRE with contrastive learning can be achieved in most languages of the XL-BEL dataset, which demonstrates that our contrastive learning framework supports other sequence-to-sequence models as well.

### 4.4.5 Case Study

To illustrate the effectiveness of the proposed model, we listed the top 1 candidates retrieved by SAPBERT$_{multi}$ and our Con2GEN in Figure 3, respectively. We can infer from Figure 3 that SAPBERT$_{multi}$ prefers entities with similar surface form as the mentions, but may miss real matching

entities. However, it clearly indicates that the proposed method can make up for this disadvantage and make comprehensive judgments by leveraging the contextual and language information of entities and mentions.

## 5 Conclusion

In this paper, we focus on both the indirect and the uninformative challenges in multilingual biomedical entity linking. Towards this, we propose Con2GEN, a prompt-based controllable contrastive generation method. Our approach outperforms baselines on 3 benchmark datasets, including XL-BEL, Mantra GSC EMEA, and Mantra GSC Patent. Further experiments and analysis demonstrate that the controllable contrastive decoding strategy of our model could considerably reduce ambiguity and meanwhile gain high performance.

## Limitations

One limitation of this work is that the generative model used in this paper requires a large amount of training data and is computationally expensive. Furthermore, due to the lack of multilingual domain-specific training data, we utilize English text and a multilingual knowledge base to construct the multilingual contextual training data. However, the constructed data may be biased for two reasons. Firstly, the number of non-English entities in the multilingual knowledge base is small, which limits the linguistic diversity of generated data. Secondly, we only replace the English mentions with non-English synonyms, and the remaining context remains in English.

## Acknowledgments

This work was supported in part by the National Key Research and Development Program of China (No. 2022ZD0116002), National Natural Science Foundation of China (No. 62006061, 61872113, 62106115, 62276075, 62106114), Science and Technology Planning Project of Shenzhen (No. JCYJ20190806112210067), and Guangdong Provincial Key Laboratory of Novel Security Intelligence Technologies, China (No. 2022B1212010005).

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

# A  Appendix

## A.1  Mapping Output to CUIs

SAPBERT$_{multi}$ only considers surface form matching and therefore cannot disambiguate the same entity name with different CUIs. However, we utilize multidimensional information, and ambiguity has

been greatly reduced. Moreover, if the gold entity is one of the matching entities, SAPBERT$_{multi}$ will consider it correct, but we randomly select one CUI from them for evaluation. Thus, if we adopt the evaluation method of SAPBERT$_{multi}$, the accuracy scores will improve than what we reported.

## A.2 Example of Trie Tree Search

For ease of understanding, we give an example of a complete decoding process by executing a trie tree search in Figure 4. For example, there is a generation path "ROOT-炭酸-of type-inorganic chemical-in-Japanese" in Figure 4. Correspondingly, there must be a record like "Name: 炭酸, Language: Japanese, Type: inorganic chemical" in UMLS. Among them, the information of name and language can be found in the UMLS file "MR-CONSO.RRF", and the information of type can be found in the UMLS file "MRSTY.RRF". We only wrote a data processing code to integrate multi-dimensional information without modification.

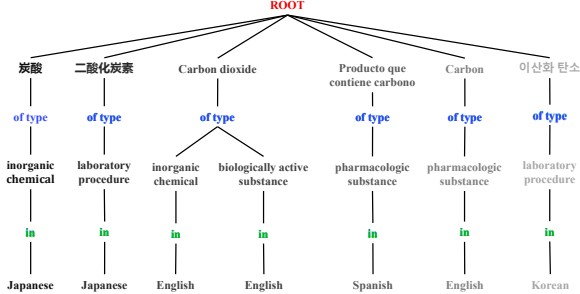

Figure 4: The trie tree of the controllable decoding algorithm for entity generation.

## A.3 Descriptions of Baselines

For the evaluation of the proposed model, we use the following SOTA baseline methods for comparison.

- **mBERT** (Devlin et al., 2019) is a multilingual contextualized language representation encoder that is pre-trained using bidirectional transformers. Following the previous work (Liu et al., 2021b), we use the pre-trained mBERT to compute vector representations. Specifically, [CLS] of the last layer's output is used as the final representation. At inference, given a query representation, a nearest neighbor search is used to rank all candidates' representations.

- **XLM-R** (Conneau et al., 2020) is a transformer-based multilingual masked language model pre-trained on text in 100 languages. The use of XLM-R in the experiments is the same as mBERT.

- **mBART** (Liu et al., 2020) is a multilingual sequence-to-sequence pre-training method that produces significant performance gains across a wide variety of tasks. We didn't finetune it on the training set.

- **BioPRO** (Zhu et al., 2022) proposes a two-stage linking algorithm to enhance the monolingual entity representations based on prompt learning. Since BioPRO is a context-infused method in the monolingual BEL task, we finetuned it on the multilingual training data.

- **SAPBERT$_{multi}$** (Liu et al., 2021b) proposes a multilingual extension of the self-alignment pre-training technique to improve domain-specialized representations in resource-lean languages. SAPBERT$_{multi}$ restricted the target UMLS ontology to only include CUIs that appear in WikiMed ($62,531$ CUIs, $399,931$ entity names), which greatly reduced the knowledge base size and task difficulty. However, we utilized the 2020AA full version of UMLS for evaluation ($4,281,184$ CUIs, $12,671,936$ entity names), which is much harder. The results in Table 2 are produced using the officially released code of SAPBERT$_{multi}$.

- **CODER** (Yuan et al., 2022b) proposes a pre-training method to use both synonyms and relations from the UMLS to direct the generation of multilingual biomedical term embeddings.

- **mGENRE** (Cao et al., 2022) proposes an autoregressive formulation to the multilingual entity linking problem in the general domain. We compared with both the base and finetuned results of mGENRE in Table 2.