# OpenReview forum: "Controllable Contrastive Generation for Multilingual Biomedical Entity Linking"
_EMNLP/2023/Conference — EMNLP 2023 Main_

### Official Review · Reviewer_iTH6 · 2023-08-04

**Soundness:** 3

**Excitement:**

3: Ambivalent: It has merits (e.g., it reports state-of-the-art results, the idea is nice), but there are key weaknesses (e.g., it describes incremental work), and it can significantly benefit from another round of revision. However, I won't object to accepting it if my co-reviewers champion it.

**Missing References:**

Yuan, H., Yuan, Z. and Yu, S., 2022, July. Generative Biomedical Entity Linking via Knowledge Base-Guided Pre-training and Synonyms-Aware Fine-tuning. In Proceedings of the 2022 Conference of the North American Chapter of the Association for Computational Linguistics: Human Language Technologies (pp. 4038-4048).

**Paper Topic And Main Contributions:**

The paper frames the MBEL task as a generative task opposed to a discriminative task. Authors compare the performance of their Con2GEn approach with SOTA with ablation studies and show improved performances. Paper also discusses about improvements achieved through contrastive learning and multidimensional information injection.

**Questions For The Authors:**

A. How is your work different from the work from Yuan, H., Yuan, Z. and Yu, S., 2022, July. Generative Biomedical Entity Linking via Knowledge Base-Guided Pre-training and Synonyms-Aware Fine-tuning?

B. Are indirect and uninformative introduces by authors?

C. What is the need to combine both the constative loss and NLL?

D. Accuracy of mGENRE is compared for several cases staring from base to +finetune to +finetune+CL, why not +finetune+type+CL?

**Reasons To Accept:**

The experiments are thorough.

**Reasons To Reject:**

-Authors refer to why greedy algorithms limitations but the alternative they use (Trie tree based decoding) is not explained clearly. E.g., -UMLS is not originally structured as you depict. Did you have to do any modifications?
-What is the need to combine both the constative loss and NLL?
-Are indirect and uninformative introduces by authors? Not convinced that they are two different drawbacks.
-Generative entity linking is not a novel concept in Biomedical entity linking, so the motivation is for multilingual aspect of BEL, but this is not adequately addressed in the paper.

**Reproducibility:**

4: Could mostly reproduce the results, but there may be some variation because of sample variance or minor variations in their interpretation of the protocol or method.

**Reviewer Confidence:**

4: Quite sure. I tried to check the important points carefully. It's unlikely, though conceivable, that I missed something that should affect my ratings.

**Typos Grammar Style And Presentation Improvements:**

There were several instances where strong claims were not supported by citations (e.g., claim starting in line 098, claim starting at line 041, )
Cite InfoNCE.
The main figure (2) is not illustrative enough to understand how the UMLS is utilized in the process, how the Trie tree decoding is utilized etc.
A through revision is needed to make the paper more readable.

---

> ### Author Rebuttal · Authors · 2023-08-29
>
> **(R denotes reasons to reject, Q denotes questions for the authors, and T denotes typos grammar style and presentation improvements)**
>
> **R1.** We gave an example of a complete decoding process by executing a trie tree search in Appendix A.1. For example, there is a generation path “ROOT-炭酸-of type-inorganic chemical-in-Japanese” in Table 4. Correspondingly, there must be a record like “Name: 炭酸, Language: Japanese, Type: inorganic chemical” in UMLS. Among them, the information of name and language can be found in the UMLS file “MRCONSO.RRF”, and the information of type can be found in the UMLS file “MRSTY.RRF”. We only wrote a data processing code to integrate multi-dimensional information without modification. Moreover, we added an ablation experiment of Con2GEN without using a trie-based controllable decoding algorithm, and the performance dropped significantly, which demonstrates the effectiveness of the proposed trie-based decoding. The results are listed as follows: Accuracy on XL-BEL dataset: EN: 52.7, ES: 4.5, DE: 5.0, FI: 2.0, RU: 3.3, TR: 7.1, KO: 5.3, ZH: 6.4, JA: 12.5, TH: 4.6. Accuracy on EMEA dataset: ES: 13.2, FR: 20.0, NL: 17.1, DE: 15.1. Accuracy on Patent dataset: FR: 23.4, DE: 20.7. The average accuracy point across all subsets is 13.3.
>
> **R2. & QC.** By combining the contrastive loss and NLL in the training phase, the Con2GEN model can be jointly optimized in a multi-task learning manner, which will improve the generalization ability of the model. Moreover, we explored different ways to combine the NLL and contrastive loss. We utilized a hyper-parameter “label weight” to balance the NLL and contrastive loss, and did a grid search from 0 to 1 with a step size of 0.1. And we found when the label weight equals 0.5, the performance is best on the validation set.
>
> **R3. & QB.** “Indirect” and “Uninformative” are different drawbacks, which are introduced in lines 66-75. Specifically, “Indirect” denotes discriminative classifiers need to iteratively fit atomic labels to learn aligned features, while generative models can directly align input and output spaces. “Uninformative” denotes ignoring the multi-dimensional information of entities, such as language, type, and context.
>
> **R4.** It is true that generative models are not a novel concept in BEL, however, multilingual BEL is much more challenging due to the lack of training data and the need for disambiguation across languages. Thus, in this paper, we not only simply apply the generative models to the MBEL task. Firstly, we propose a novel prompt-based controllable contrastive decoding algorithm, which can guide the generation process using a predefined natural language template and contrastive learning. In this way, the multidimensional knowledge of biomedical entities can be injected and exploited during inference. Secondly, we alleviate the problem of lacking MBEL training data by proposing a weakly supervised data augmentation method. Finally, extensive experiments show that our model achieves promising performance improvements spanning 12 typologically diverse languages.
>
> **QA.** The differences between our work and the paper “Generative Biomedical Entity Linking via Knowledge Base-Guided Pre-training and Synonyms-Aware Fine-tuning” are as follows: (1) Different decoding objectives: Yuan. et al., 2022 only generate the entity name at the decoding phase, ignoring the multidimensional information of bio-entities and increasing the difficulty of disambiguation. In our paper, we proposed a constrained decoding algorithm consisting of a predefined template for multidimensional information injection during inference. (2) Different training manners: Yuan. et al., 2022 pre-trained the model to improve the generalization ability. Instead, we utilized multi-task learning to jointly optimize the contrastive loss and the task loss and meanwhile to strengthen the proposed Con2GEN model.
>
> **QD.** In order to analyze the contribution of “type” and “CL” separately, we reported the performances of “+finetune+type” and “+finetune+CL” methods in Table 3, respectively. As suggested, we have added the experiment of “+finetune+type+CL”, and the accuracy on each language of the XL-BEL dataset is as follows:
> | mGENRE | EN | ES | DE | FI | RU | TR | KO | ZH | JA | TH | Avg |
> |-------------------|-------|------|------|------|------|------|------|------|------|------|------|
> | +finetune+type+CL         | 86.8  | 50.9 | 30.7 | 24.8 | 41.9 | 40.2 |16.9 | 16.7 | 24.9 | 18.3 | 35.2 |
>
> **T.** Thank you for pointing out the flaws in our writing, we will carefully proofread our manuscript in the next revision.

---

### Official Review · Reviewer_LBDQ · 2023-08-08

**Soundness:** 4
**Typos Grammar Style And Presentation Improvements:** 1) section 3.3 could be removed or sh…

**Excitement:**

4: Strong: This paper deepens the understanding of some phenomenon or lowers the barriers to an existing research direction.

**Paper Topic And Main Contributions:**

This paper presents a new approach for multilingual biomedical entity linking (MBEL); instead of using the classifier-based approach or vector space model, they formulate the task as a sequence-to-sequence generation task, and propose a prompt-based controllable contrastive generation framework. They design a prompt template to encourage the model to learn both semantic type and language information about the entity name; their controllable decoding algorithms reduces the output space thus improving the computational cost during inference; their contrastive training objective could  learn more discriminative information about hard negative examples. They evaluate their approach on multiple datasets including 1000 test examples for each one of the 10 languages from XL-BEL dataset, and a few 100 examples from EMEA and patent dataset, their approach achieves better performances over a few baseline models.

**Questions For The Authors:**

1) What are the statistics for the training set? Why only use the smaller subset for evaluation?
2) How do you swap english mentions and non-english entities of the data? Any translator in this process?
3) I would expect more details on each baseline, like how do you use mBERT or XLM-R for the task, multiclass classification or just using them to compute vector representations? What are their output spaces and what are the sizes of their training data if it is trained?
4) It seems mGENRE is not trained at table 1, are other models also not trained? What are the fine-tuning performances of mGENRE on all datasets? Are they better than the fine-tuned Con2Gen?
5) What are the performances of models without using constraint decoding algorithms?
6) Have you explored different ways to combine the NLL and contrastive loss?
7) How do you map the output of the sequence-to-sequence model to CUIs? Is it 1-to-1 mapping?

**Reasons To Accept:**

This paper combines a few techniques (prompt based controllable decoding, and contrastive learning for text generation) together for the MBEL task, which is a pretty important task in the biomedical domain. Their approach achieves better performances than a few baseline approaches. And they also conduct a few ablation studies to understand the contribution of each component.

**Reasons To Reject:**

I have a few main concerns about their paper: 1) for each language from the XL-BEL dataset, only 1000 instances are selected from each dataset, and also a few hundreds examples from EMEA and patent dataset, why only use such smaller subset for evaluation, not the entire dataset from XL-BEL? Also more statistics about the training and validation data. 2) it’s unclear whether the performances are significant or not? Especially for the model without using contrastive learning, it seems only 0.2 percent lower than the complete model, which does not support their claim of the contrastive learning, and it seems their ideas are from the paper “Contrastive Neural Text Generation”, which does not bring new things to the reader. 3) their controllable decoding algorithm is very similar to the work of GENRE, they ignore to mention this part in the related work and method section. Also, it seems fine-tuning mGENRE on the same data may achieve better performances, but the authors do not evaluate all the data and compare it fairly. 4) There are also a few ablation studies I expect to see, like the performances of models without using controllable decoding, and different hard examples for contrastive learning. 5) The output of the sequence-to-sequence model is the entity text plus its type and language information,however the evaluation target is the CUI. Mapping the output to CUI is not a trivial task, but authors did not mention anything for this. In general, I like the task of this paper and the idea of combining different techniques, but I would expect more details for points 1) - 5).

**Reproducibility:**

3: Could reproduce the results with some difficulty. The settings of parameters are underspecified or subjectively determined; the training/evaluation data are not widely available.

**Reviewer Confidence:**

4: Quite sure. I tried to check the important points carefully. It's unlikely, though conceivable, that I missed something that should affect my ratings.

---

> ### Author Rebuttal · Authors · 2023-08-29
>
> **(R denotes reasons to reject, Q denotes questions for the authors, and T denotes typos grammar style and presentation improvements)**
>
> **R1. & Q1.** For the sake of fairness, we follow the same test sets as SAPBERTmulti (Liu et al., 2021b) and CODER (Yuan et al., 2022). Since SAPBERTmulti (Liu et al., 2021b) and CODER (Yuan et al., 2022) used smaller subsets for evaluation, we are consistent with these existing publications. We briefly describe the statistics of the training and validation sets in lines 446-448. Specifically, more statistics about the training and validation sets are as follows: For the training set, the number of sentences is 30,349, the number of mentions is 3,136,568, and the number of entities is 25,529. For the validation set, the number of sentences is 10,058, the number of mentions is 412,905, and the number of entities is 12,535.
>
> **R2.** We believe the improvements are significant since we only trained one single model which outperforms baselines on almost all languages. Although the contrastive learning strategy improved the average accuracy by only 0.2 points, it worked across 9 diverse languages, which is not trivial. Moreover, we can infer from Table 3 that the proposed contrastive learning strategy also makes mGENRE work better, demonstrating its generalization. Although the paper “Contrastive Neural Text Generation” also utilizes contrastive learning, our training objectives are quite different from theirs. Concretely, for each instance, we constructed “semantically confusing negatives” and “hard negatives” specifically for the MBEL task. Moreover, when computing the contrastive loss, we only utilized the representations of the decoder since the representations of the decoder are the same, however, the paper “Contrastive Neural Text Generation” utilized both the representations of the source and the representations of the target.
>
> **R3.** Since mGENRE is the extended version of GENRE in multilingual scenarios, we only cited mGENRE in our manuscript. Thank you for pointing this out, we will cite GENRE in the next revision. For the sake of fairness, we have added experiments of finetuning mGENRE on EMEA and Patent datasets. The fine-tuning performances of mGENRE on all datasets are as follows: Accuracy on XL-BEL dataset: EN: 85.4, ES: 49.8, DE: 31.3, FI: 24.8, RU: 41.5, TR: 39.5, KO: 16.0, ZH: 16.9, JA: 23.9, TH: 17.2. Accuracy on EMEA dataset: ES: 62.0, FR: 56.4, NL: 55.1, DE: 54.1. Accuracy on Patent dataset: FR: 61.2, DE: 55.7. The average accuracy of finetuned mGNERE is 85.4, which is worse than our Con2GEN.
>
> **R4. & Q5.** As suggested, we have added the ablation study of Con2GEN without using controllable decoding. The experimental results are as follows: Accuracy on XL-BEL dataset: EN: 52.7, ES: 4.5, DE: 5.0, FI: 2.0, RU: 3.3, TR: 7.1, KO: 5.3, ZH: 6.4, JA: 12.5, TH: 4.6. Accuracy on EMEA dataset: ES: 13.2, FR: 20.0, NL: 17.1, DE: 15.1. Accuracy on Patent dataset: FR: 23.4, DE: 20.7. The average accuracy point across all target languages is 13.3, which dropped significantly compared to Con2GEN, demonstrating the effectiveness of the proposed controllable decoding.
>
> **R5. & Q7.** It is true that mapping the output to CUI is a non-trivial task. We mentioned in lines 234-242 that in the general domain the Wikipedia title can be used to uniquely identify an entity, however, in the biomedical domain, bio-entity names with type and language information are still ambiguous combinations. However, by adding multi-dimensional information, ambiguity has been greatly reduced. In the experiment, if the combination of entity name, type, and language cannot uniquely identify a CUI, we will randomly select one CUI from them. Moreover, if we combine these CUIs, the performance will be further improved.
>
> **Q2.** We propose a straightforward idea of swapping the English mention of input with the multilingual entity of output. Since the Medmentions dataset only contains English instances, we use this simple method to generate multilingual input. We didn’t use any translator in this process, which will not introduce any extra noise. Moreover, we discuss the possible limitations of our constructed data in lines 592-603.
>
> **Q3.** Following the previous work (SAPBERTmulti (Liu et al., 2021b)), we use these pre-trained baselines to compute vector representations. Specifically, [CLS] of the last layer’s output is used as the final representation. At inference, given a query representation, a nearest neighbor search is used to rank all candidates’ representations.
>
> **Q4.** Following the previous work (SAPBERTmulti (Liu et al., 2021b)), we also didn’t train other models. The fine-tuning performances of mGENRE on all datasets are as follows: Accuracy on XL-BEL dataset: EN: 85.4, ES: 49.8, DE: 31.3, FI: 24.8, RU: 41.5, TR: 39.5, KO: 16.0, ZH: 16.9, JA: 23.9, TH: 17.2. Accuracy on EMEA dataset: ES: 62.0, FR: 56.4, NL: 55.1, DE: 54.1. Accuracy on Patent dataset: FR: 61.2, DE: 55.7. The average accuracy of finetuned mGNERE is 85.4, which is worse than our Con2GEN.
>
> **Q6.** We have explored different ways to combine the NLL and contrastive loss. Specifically, we utilized a hyper-parameter “label weight” to balance the NLL and contrastive loss, and did a grid search from 0 to 1 with a step size of 0.1. We found when the label weight equals 0.5, the performance is best on the validation set. Due to the length limit, we didn’t mention it in the manuscript.
>
> **T.** Thank you for pointing out the flaws in our writing, we will carefully proofread our manuscript in the next revision.

---

### Official Review · Reviewer_F782 · 2023-08-10

**Soundness:** 3

**Excitement:**

4: Strong: This paper deepens the understanding of some phenomenon or lowers the barriers to an existing research direction.

**Paper Topic And Main Contributions:**

The authors propose a framework for multilingual biomedical entity linking that focuses on generating natural language sentences summarizing information from UMLS concepts in biomedical texts. Their approach utilizes a sequence-to-sequence generation task and is designed to match UMLS concepts across various languages and types, facilitating the disambiguation of cross-information.

**Questions For The Authors:**

1. I am not clear what the authors mean by 'fails to capture the interactions between them' (line 68) - what interactions are you referring to?

2. I do not believe this is completely true about current work only solving the MBEL task via shallow matching on surface forms ignoring the context. Please see: Wu et al 2019 (https://aclanthology.org/2020.emnlp-main.519/)



**Reasons To Accept:**

1. Explores biomedical entity linking across 12 different languages
2. Conduct an evaluation across three different datasets
3. Interesting approach that leverages summarising text into a predefined template to generate the entities.


**Reasons To Reject:**

I really liked the approach, however:

1 the authors disregard previous work that has incorporated context which needs to be addressed

2. it is not clear to me how all the baseline comparisons were conducted (e.g. what is the mBERT method in Table 1 - the appendix only describes the language model itself - not how the classification was conducted)

3. it is also not clear where the accuracy of all of the results is coming from (e.g. SAPBERTmulti Liu et al 2021b - where did the 86.2 in EN XL-BEL come from - I don't see that result in the cited paper)

**Reproducibility:**

3: Could reproduce the results with some difficulty. The settings of parameters are underspecified or subjectively determined; the training/evaluation data are not widely available.

**Reviewer Confidence:**

3: Pretty sure, but there's a chance I missed something. Although I have a good feel for this area in general, I did not carefully check the paper's details, e.g., the math, experimental design, or novelty.

---

> ### Author Rebuttal · Authors · 2023-08-29
>
> **(R denotes reasons to reject and Q denotes questions for the authors)**
>
> **R1. & Q2.** Wu et al., 2019 propose the BLINK model to solve the monolingual entity linking problem in the general domain, which also considers contextual information. However, monolingual entity linking approaches cannot be effectively applied to other languages due to the huge discrepancies between multilingual and monolingual versions of the entity linking task. Note that in Table 1, we also reported the results of BioPRO (Zhu et al., 2022), which is a SOTA method in the monolingual BEL task and incorporates context. Thank you for pointing out these citations, we will carefully proofread our manuscript in the next revision.
>
> **R2.** Following the previous work (SAPBERTmulti (Liu et al., 2021b)), we use these pre-trained baselines to compute vector representations. Specifically, [CLS] of the last layer’s output is used as the final representation. At inference, given a query representation, a nearest neighbor search is used to rank all candidates’ representations.
>
> **R3.** SAPBERTmulti (Liu et al., 2021b) restricted the target UMLS ontology to only include CUIs that appear in WikiMed (62,531 CUIs, 399,931 entity names), which greatly reduced the knowledge base size and task difficulty. However, we utilized the 2020AA full version of UMLS for evaluation (4,281,184 CUIs, 12,671,936 entity names), which is much harder. The results in Table 1 are produced using the officially released code of SAPBERTmulti (Liu et al., 2021b).
>
> **Q1.** Our intention is that discriminative classifiers need to iteratively fit atomic labels to learn aligned features, which is an indirect manner. However, generative models inherently have a better ability to align input and output spaces than discriminative models, which will better exploit the shared dependencies (interactions) between source mentions and target entities.

---

### Official Review · Reviewer_4M3G · 2023-08-12

**Soundness:** 3

**Ethical Concerns:**

Yes

**Excitement:**

3: Ambivalent: It has merits (e.g., it reports state-of-the-art results, the idea is nice), but there are key weaknesses (e.g., it describes incremental work), and it can significantly benefit from another round of revision. However, I won't object to accepting it if my co-reviewers champion it.

**Paper Topic And Main Contributions:**

The paper studies the problem of multilingual biomedical entity linking, which is essential for various downstream applications. Specifically, the paper proposes a prompt-based controllable contrastive generation framework by summarizing the UMLS concept's multidimensional information mentioned in biomedical text into a natural sentence following a predefined template.
Furthermore, the proposed method matches against UMLS concepts in as many languages and types as possible, facilitating cross-information disambiguation.
Experimental results show that the proposed algorithm is competitive with SOTA baselines.

**Reasons To Accept:**

The paper addresses a significant problem by proposing a prompt-based controllable contrastive generation framework, and several improvement techniques have been designed. The experimental evaluation shows that the proposed algorithm improves the preexisting algorithms.

**Reasons To Reject:**

The main innovative ideas are oriented from preexisting works in NLP, wherein commonly adopted. LLM, Multi-Prompt, and Curriculum Learning might further improve the optimization technique via language awareness. Moreover, there are significant differences between the general and medical domains, and the domain migration strategy might significantly impact the model performance. Therefore, they should conduct more comprehensive experiments to prove the algorithm's performance and make the experimental results more convincing.

**Reproducibility:**

3: Could reproduce the results with some difficulty. The settings of parameters are underspecified or subjectively determined; the training/evaluation data are not widely available.

**Reviewer Confidence:**

4: Quite sure. I tried to check the important points carefully. It's unlikely, though conceivable, that I missed something that should affect my ratings.

---

> ### Author Rebuttal · Authors · 2023-08-29
>
> **(R denotes reasons to reject)**
>
> **R.** It is true that the main innovative ideas are oriented from preexisting works. However, multilingual BEL is a challenging task due to the lack of training data and the need for disambiguation across languages. Thus, in this paper, we not only simply apply preexisting approaches to the MBEL task. Firstly, we propose a novel prompt-based controllable contrastive decoding algorithm, which can guide the generation process using a predefined natural language template and contrastive learning. In this way, the multidimensional knowledge of biomedical entities can be injected and exploited during inference. Secondly, we alleviate the problem of lacking MBEL training data by proposing a weakly supervised data augmentation method. Finally, extensive experiments show that our model achieves promising performance improvements spanning 12 typologically diverse languages.
>
> As suggested, we have conducted more comprehensive experiments to prove the algorithm's performance and make the experimental results more convincing.
>
> Firstly, we added the ablation study of Con2GEN without using controllable decoding. The experimental results are as follows: Accuracy on XL-BEL dataset: EN: 52.7, ES: 4.5, DE: 5.0, FI: 2.0, RU: 3.3, TR: 7.1, KO: 5.3, ZH: 6.4, JA: 12.5, TH: 4.6. Accuracy on EMEA dataset: ES: 13.2, FR: 20.0, NL: 17.1, DE: 15.1. Accuracy on Patent dataset: FR: 23.4, DE: 20.7. The average accuracy point across all target languages is 13.3, which dropped significantly compared to Con2GEN, demonstrating the effectiveness of the proposed controllable decoding.
>
> Secondly, we added experiments of finetuning mGENRE on EMEA and Patent datasets. The fine-tuning performances of mGENRE on all datasets are as follows: Accuracy on XL-BEL dataset: EN: 85.4, ES: 49.8, DE: 31.3, FI: 24.8, RU: 41.5, TR: 39.5, KO: 16.0, ZH: 16.9, JA: 23.9, TH: 17.2. Accuracy on EMEA dataset: ES: 62.0, FR: 56.4, NL: 55.1, DE: 54.1. Accuracy on Patent dataset: FR: 61.2, DE: 55.7. The average accuracy of finetuned mGNERE is 85.4, which is worse than our Con2GEN.
>
> Finally, we added the experiment of “mGENRE+finetune+type+CL”, and the accuracy on each language of the XL-BEL dataset is as follows: EN: 86.8, ES: 50.9, DE: 30.7, FI: 24.8, RU: 41.9, TR: 40.2, KO: 16.9, ZH: 16.7, JA: 24.9, TH: 18.3, Avg: 35.2. Our proposed Con2GEN still outperforms the method mGENRE+finetune+type+CL.

---

### Meta-Review · Area_Chair_4pLN · 2023-09-14

**Recommendation:** 4

**Metareview:**

There is a consensus among the reviewers and most questions have been answered in the author response.

---

### Decision · Program_Chairs · 2023-10-07

**Decision:**

Accept-Main

**Comment:**

There is a consensus among the reviewers and most questions have been answered in the author response.